# On Your Mark, Get Set, Self-Control, Go: A Differentiated View on the Cortical Hemodynamics of Self-Control during Sprint Start

**DOI:** 10.3390/brainsci10080494

**Published:** 2020-07-29

**Authors:** Kim-Marie Stadler, Wanja Wolff, Julia Schüler

**Affiliations:** 1Department of Sport Science, University of Konstanz, 78464 Konstanz, Germany; kim-marie.stadler@uni-konstanz.de (K.-M.S.); wanja.wolff@uni-konstanz.de (W.W.); 2Department of Educational Psychology, University of Bern, 3012 Bern, Switzerland

**Keywords:** sprint start, self-control, cerebral oxygenation, ventral-lateral-prefrontal-cortex

## Abstract

Most sports are self-control demanding. For example, during a sprint start, athletes have to respond as fast as possible to the start signal (action initiation) while suppressing the urge to start too early (action inhibition). Here, we examined the cortical hemodynamic response to these demands by measuring activity in the two lateral prefrontal cortices (lPFC), a central area for self-control processes. We analyzed activity within subregions of the lPFC, while subjects performed a sprint start, and we assessed if activation varied as a function of hemisphere and gender. In a counterbalanced within-subject design, 39 participants (age: mean (M) = 22.44, standard deviation (SD) = 5.28, 22 women) completed four sprint start conditions (blocks). In each block, participants focused on inhibition (avoid false start), initiation (start fast), no start (do not start) and a combined condition (start fast; avoid false start). We show that oxyhemoglobin in the lPFC increased after the set signal and this increase did not differ between experimental conditions. Increased activation was primarily observed in ventral areas of the lPFC, but only in males, and this increase did not vary between hemispheres. This study provides further support for the involvement of the ventral lPFC during a sprint start, while highlighting gender differences in the processing of sprint start-induced self-control demands.

## 1. Introduction

Winning or losing in a sprint race is determined by milliseconds and is highly dependent on a perfect start [1,2,3,4]. The sprint start contributes 5% of the overall 100 m race time [5]. Also, in other sports, the start plays a decisive role for the overall performance. For example, in the shortest swimming races (50 m), the start accounts for 33% of the total race time, making it even more important for the overall result [6]. Taken together, in many different sports, a fast and excellent race start plays a central role in determining the race outcome [7,8,9].

Start performance hinges on a diverse set of variables. For example, start position [5,10], force rate [2,11], reaction time and acceleration [4,12], they all affect how well an athlete is able to start. However, cognitive processes also contribute to a good sprint start. For example, an external attentional focus has been shown to be more effective for the start reaction time than an internal attentional focus [13]. Moreover, agility (i.e., the capacity for rapid movement changes during a sprint) is dependent on effective information processing [13,14]. To illustrate, during a sprint start, the athlete needs to recognize and process the acoustic start signal, make the decision to initiate the start and enact this decision by producing the required movement. Thus, the athlete is required to inhibit the urge to start until a signal indicates otherwise (i.e., action inhibition) and then initiate a movement as fast as possible (action initiation).

Action inhibition and initiation are two processes that rely on self-control [15,16]. Self-control is defined as the ability to regulate and alter “one’s own inner states, processes and responses (..), including (..) actions, thoughts, feelings, desires and performances” [17] (pp. 6–7) to achieve a specific goal. Self-control is conducive to a plethora of positive short- and long-term outcomes [18,19,20,21].

Exerting self-control is, however, not easy, and is associated with negative experiences. People often fail to apply the amount of self-control that would be required [20]. For example, we often do not manage to resist a short-term temptation (e.g., snacking) that would be at odds with a long-term goal (lose weight). Exerting self-control is experienced as effortful and aversive [22,23,24,25] and therefore, people tend to avoid exerting self-control if possible [24]. In line with this, most theories on self-control suggest that applying self-control produces costs, and is therefore invested sparingly [22,23,24,26]. For example, it has been suggested that the effort one experiences while applying self-control serves as a signal to indicate the costs of one’s current action and serves as a prompt to withhold further effort [26]. In line with this, a large body of research on the ego-depletion effect has shown that prior self-control exertion leads to impaired subsequent performance in non-sporting [27] as well as in sporting tasks [28,29]. For example, Englert and Bertrams [30] showed that participants who had completed a self-control-demanding task prior to a basketball free-throw test performed worse than participants who had not worked on a self-control-demanding task. More importantly for the present paper, research has shown that participants who had completed a self-control-demanding task prior to a sprint start displayed worse reaction times and more false starts in subsequent sprint starts [31,32]. In sum, a large body of research shows that self-control is important for sports performance in general [29] and emerging research has found support for the proposed importance of self-control for sprint starts, too [31,32].

## 2. Cortical Underpinnings of Self-Control

In light of its importance for goal-directed behavior, a large body of neuroscientific research has focused on understanding the neuroscience of self-control [24,33,34,35]. Much evidence points to an executive control network that governs self-controlled, top-down processing [24,34,35,36]. According to this approach, self-control can be broken down into the three components, namely, specification, regulation and monitoring [37]. Thus, self-control consists of a continuous loop in which control signals are specified, applied and the outcome is monitored to assess whether the control signal needs to be adjusted. It has been proposed that the specification and monitoring components of control are performed by the dorsal Anterior Cingulate Cortex (dACC), whereas regulation is primarily executed by the lateral prefrontal cortex (lPFC) [37]. Thus, research suggests that the actual process of applying self-control, for example to control the impulse to start too early in a sprint race, is executed by the lPFC. Additional support for the proposed role of lPFC in self-control processes comes from clinical research, where, for example, patients who suffer from attention deficit hyperactivity disorder have been shown to display lower activation in lPFC during control-demanding tasks [38]. Depending on the specific control function that is required in a situation, different areas of the lPFC are engaged in the control process. For example, dorsolateral areas seem to govern top-down attention control, whereas ventrolateral areas seem important for response inhibition [39].

The cortical underpinnings of self-control in sports—and for sprint starts specifically—are less well understood. In line with research and theorizing from cognitive neuroscience, self-control demands in sports covary with lPFC activation [40]. While a number of studies has already investigated lPFC activity during endurance performance [41,42], only one study has assessed lPFC activation during a sprint start [43]. In a within-subject repeated measures design, male participants performed three sprint start sequences while lPFC activity was measured with functional near-infrared spectroscopy (fNIRS) [43]. A significant increase in lPFC oxygenation prior to the start signal indicated high self-control demands during a sprint start. In this study, exploratory analyses indicated that different subareas of the lPFC were active in conditions that differed in regard to the self-control demands they supposedly imposed: if participants only had to avoid producing a false start (i.e., action inhibition), this was accompanied by a strong increase in ventral parts of lPFC and a less pronounced increase in anterior parts of the lPFC, compared to when participants had to focus on avoiding a false start, while also producing a maximally fast start (action initiation). In addition, no activation changes in posterior parts of the lPFC were observed, irrespective of condition. Hence, preliminary empirical evidence supports the presumed role of lPFC during sprint starts. It is our research goal to better understand how the lPFC covaries with the self-control demands of a sprint start. Thus, the present study takes up the results of Wolff et al. [43] and tests them within the framework of a previously formulated hypothesis.

## 3. Present Study

In this study, we aim to replicate and extend prior research on the lPFC involvement during sprint starts [43]. Specifically, we use fNIRS to capture cortical oxygenation changes during a sprint start sequence that consists of the prompts “On your marks”, “Set” and “Go”. Aiming at replicating Wolff et al. [43], we test if lPFC oxygenation increases in the time interval between Set and Go. Extending their work on a general level, we assess if lPFC oxygenation increases differ as a function of participant gender. This is an important extension because research points towards sex differences in self-control performance [44,45]. In addition, some research points towards a differential involvement of left and right hemispheres in self-control [46,47]. Thus, as a second extension, we test if lPFC oxygenation changes differ between the left and the right hemisphere.

In addition, we test if the facets of self-control demands (action initiation, action inhibition), that we aim to trigger using different instructions for the sprint start, cause specific changes in the lPFC. Our different start conditions were developed in order to emphasize the self-control demands for action inhibition and action initiation: Participants either had to focus on action inhibition (avoid a false start), on a fast action initiation (start as soon as possible), on both (start as soon as possible and avoid a false start) or did not start at all (no start), which was expected to increase the demand for action inhibition. Since all conditions impose self-control demands, we test the hypothesis that lPFC activation would significantly increase in all four sprint start conditions (action inhibition, action initiation, combined, no-start). However, since the no-start condition only imposes the self-control demand to inhibit any start action on accident, we expect the lPFC activation to be the lowest in this condition.

Finally, we a priori tested the activation differences of specific regions of the lPFC to follow up on the explorative findings by Wolff et al. [43]. Specifically, Wolff et al. [43] used fNIRS to monitor oxygenation changes over different parts of the lPFC and found that specific parts of their montage according to the international 5/10 system responded with increased activation during the sprint start (anterior: F6–AF8, AF4–AF8, AF3–AF7, F5–AF7; ventral: F6–F8, F6–FC6, F5–FC5, F5–F7), whereas no activation changes were observed for the other parts (F6–F4, AF4–F4, FC4–FC6, FC4–F4, F2–F4, F1–F3, AF3–F3, FC3–F3, FC3–FC5, F5–F3). Thus, we test the hypothesis that ventral and anterior parts of the lPFC show significantly higher activation in all four start conditions than the remaining parts of the chosen lPFC montage.

## 4. Methods

### 4.1. Design

The study was conducted in the Sport Psychology Lab of the authors´ University and was based on an experimental, randomized within-subject design. Thus, each subject participated in all four start conditions.

### 4.2. Participants

We recruited a sample of *n* = 60 participants (27 male, 33 female; mean (M) M*_age_* = 22.44, standard deviation (SD) SD*_age_* = 5.28). The majority of the participants studied psychology (*n* = 33) or sport science (*n* = 6). The requirements for study participation were German language skills, age between 18 and 30 years to minimize age-dependent differences in cortical oxygenation and no extensive experience with sprint starts but doing sport regularly. Participants reported to be physically active for three to four times a week (M_weeklysport_ = 3.25, SD_weeklysport_ = 2.11), with one exercise session lasting on average 67.67 min (SD_minutes_ = 31.64). The main sport activities were running, weight-lifting and dancing. They received 15 EUR for participation. Due to recording errors and missing trigger definitions in the raw data, only *n* = 39 (22 women) of the full dataset remained for statistical analyses. The study was conducted in agreement with the declaration of Helsinki and does not fall within the remit of the Ethics Committee of the University.

### 4.3. Procedure

The testing session lasted approximately 90 min and study sessions were conducted whenever participants were available. Participants were welcomed, received and signed a written informed consent form and completed a demographic questionnaire. They were asked whether they followed the instructions (no caffeine and alcohol consumption and no exercise 24 h prior to the experiment, and no caffeine two hours before) as requested prior to the testing session. The experimenter explained the task and all four start conditions carefully. Sprint starts were carried out in a standing position. Participants were given time to practice the sprint start sequence until they felt confident in starting with the experiment. This was not standardized but participants chose freely how much practice they needed before starting. Next, fNIRS equipment was prepared and set up. After calibrating and optimizing the signal, participants wore the fNIRS recorder in a backpack. Hemodynamic changes were recorded in real-time and transferred via Team Viewer^®^ to an external computer and signal quality was monitored constantly. Then, the sprint start procedure started.

The sprint start sequence was the same as in the study of Wolff et al. [43]. The same consecutive start signals (“On your marks”, “Set”, “Go”) were vocally presented to the participants for all start conditions. The instructions for the sprint starts, which represent the experimental manipulation (see below), were presented by a computerized voice to avoid possible biases by the experimenter and to ensure the same inter stimulus interval (see Figure 1).

Participants completed four blocks (i.e., start conditions). Participants either had to focus on false start inhibition (Action inhibition start condition, “Avoid a false start!”), on action initiation (Action initiation start condition, “Start as fast as possible!”), on both (combined start condition, “Start as fast as possible and avoid a false start!”) or did not start at all (no-start condition, “Don’t start!”). Each block consisted of 10 trials. Each block started with a 60 s baseline measurement of the prefrontal activity in a standing position while participants fixated on a black cross on the floor. Then, the instructions about the upcoming start condition followed. On the signal “On your marks”, subjects placed their supporting leg behind the starting line, which was marked by black tape on the floor. The other foot was placed on a start mat (rectangular mat on the floor). On the signal “Set”, which followed six seconds later, participants bent their knees and hip to shift their weight onto the front leg. Seven seconds after the “Set” signal, the “Start” signal was given, and in the action inhibition, action initiation and combined start condition participants started. In the no-start condition, no response was required. Subjects ended their sprint at a “stop” line five meters after the start line. Each trial was followed by a 30 s break to ensure that oxygenation changes returned to baseline level [48]. The order of conditions was counterbalanced to prevent order effects. After the experiment, participants were asked to answer follow-up questions (motivation, perceived task difficulty, momentary exhaustion). Finally, they were debriefed, paid and thanked for their participation.

### 4.4. Measures

fNIRS measurement: A multichannel continuous wave fNIRS imaging system (NIRSport, NIRx Medical Technologies LLC, Los Angeles, NY, USA) was used to measure hemodynamic changes during the sprint start. fNIRS is non-invasive and can be used during active sport exercises (e.g., cycle ergometry, treadmill walking, running) [49,50,51,52]. fNIRS measures changes in oxygenated (HbO) and de-oxygenated (HbR) hemoglobin in the cerebral cortex using NIR light [53]. 8 × 8 (8 Sources + 8 Detectors) optodes were placed bilaterally (two 4 sources + 4 detectors) according to the international 5/10 system [54]. We used the same montage of optodes as described in Wolff et al. [43] (Figure 2).

The cap was set up and fixed so that CV-point 14 was placed in the middle of the head (half-length distance between both ears and half-length distance between nasion and inion). Furthermore, an overcap was put over the probe holder cap to secure and improve optode contact and to minimize the impact from ambient light.

Hemodynamic changes in the lPFC were measured 2 s before the set signal (baseline) and 5 to 7 s after the set signal (response to set signal).

Follow-up questions: On a 7-point Likert scale (1 = not at all, 7 = very much), participant’s motivation to execute the sprint starts as fast as possible was assessed. In addition, participants were asked to indicate (open answer field) which sprint start condition was the most difficult for them (no-start, action inhibition start, action initiation start, combined start). Finally, momentary exhaustion was assessed on a 7-point Likert scale (1 = not at all, 7 = very much): “How exhausted do you fell right now?”

### 4.5. Preprocessing

fNIRS preprocessing: To preprocess raw fNIRS data, Homer2 was used [55]. First, channels (source-detector combinations) with too high or too low optical density were removed using the enPruneChannels function with the following function arguments: d*Range*(1) = 1e^−2^; d*Range*(2) = 3e; SNR*thresh* = 2; SD*range*(1) = 0.0; SD*range*(2) = 45.0, *reset* = 0. Secondly, by taking the logarithm of the signal, raw optical intensity data was converted into changes in optical density (OD). To correct motion artifacts, we used the Wavelet_Motion_Correction function with an IQR of 1.5, which is known to be efficient in recovering the hemodynamic response function [56,57,58]. Remaining motion artifacts were removed using the *hmrMotionArtifact* function with the following arguments: t*Motion* = 0.5; t*Mask* = 1.0; STDEV*thresh* = 10.0; AMP*thresh* = 1.00. If a Set or Start signal was within a time range of −3 to 10 s of a detected-motion artifact, this trial was removed from further analysis. Following Wolff et al. [43], the corrected data were low-pass-filtered using a cut-off frequency at 0.5 Hz and converted into oxygenated (HbO) and de-oxygenated (HbR) hemoglobin concentration changes using the modified Beer–Lambert law [59]. Finally, referring to Essenpreis, Cope, Elwell, Arridge, van der Zee & Delpy [60], path length factors were chosen differently for the two wavelengths (7.3 for 760 nm and 6.4 for 850 nm) and the *hmBlockAvg* function was applied to obtain corrected group averaged oxygenation values [55].

Channel selection: After preprocessing the data, time interval plots (−2 s before the Set signal to 7 s after Start signal) of averaged hemodynamic changes, for each sprint start condition, were illustrated in Homer2. Explorative findings of Wolff et al. [43] identified some channels (source–detector combinations) as relevant, as these showed a high hemodynamic response during self-control execution while other channels did not respond to the experimental demands. To test the robustness of these findings, we grouped the channels as separate regions of interest (ROIs) following Wolff et al. [43]. We grouped two anterior channels on the right hemisphere (F6–AF8, AF4–AF8) and two anterior channels on the left hemisphere (AF3–AF7, F5–AF7) as anterior parts of the lPFC. Two ventral channels on the right hemisphere (F6–F8, F6–FC6) and two ventral channels on the left hemisphere (F5–FC5, F5–F7) were grouped as ventral parts of the lPFC. The five remaining channels on the right hemisphere (F6–F4, AF4–F4, FC4–FC6, FC4–F4, F2–F4) and the five remaining channels on the left hemisphere (F1–F3, AF3–F3, FC3–F3, FC3–FC5, F5–F3) were summarized as other parts of the lPFC. Hence, we analyzed them (anterior, ventral, others) as three ROIs.

## 5. Results

### 5.1. Manipulation Check

Motivation during the experiment and fatigue level after the experiment were measured on a 7-point Likert scale (1 = not at all, 7 = very much). Participants were generally motivated to perform well (M = 5.865; SD = 0.905; Range = 4–7) and were not very exhausted directly after the experiment (M = 2.919, SD = 1.937, Range = 1–7). Contrary to our expectations, 33.3% (*n* = 12) of the participants rated the no-start condition as being the most difficult, compared with 25% (*n* = 9) who perceived the action initiation start condition, 16.7% (*n* = 6) the combined start condition and 8.3% (*n* = 3) the action inhibition start condition as the most difficult. Six participants did not answer the question, but were still included in subsequent analyses.

### 5.2. Cortical Activity during Sprint Start

Data was restructured and merged using Matlab (R2016a; Natick, Massachusetts: The MathWorks Inc.). To test if lPFC oxygenation increases in the time-interval between Set and Go, we conducted a three-way repeated measures analysis of variance (ANOVA), having three within factors: The first factor was Time (Baseline vs. Set), comparing the lPFC activation 2 s before the start (baseline) and 5 to 7 s after the set signal (response to set signal). The second factor was Condition (no-start, action inhibition start, action initiation start, combined start), and the third factor was region of interest (ROI) (Anterior, Ventral, Others). Oxygenated hemoglobin (HbO) concentration was analyzed as the dependent variable. De-oxygenated hemoglobin (HbR) concentration was not analyzed. To assess differences between factor levels, Bonferroni-corrected post-hoc *t*-tests were computed. Statistical analyses were executed in R (3.5.1; R Core Team, Vienna, Austria, 2018). The assumption of sphericity was met for all repeated measure ANOVAs. We set statistical significance at α = 0.05. We calculated partial η^2^ as effect-size estimates [61].

No significant three-way interaction between Time, Condition and ROI was found, *F* (4,168) = 0.346, *p* = 0.864, η^2^ = 0.009. A significant main effect for Time was found, *F* (1,38) = 7.290, *p* = 0.010, η^2^ = 0.161. Hence, oxygenated hemoglobin significantly increased from 2 s before the Set signal to 7 s after the Set signal. Furthermore, a significant main effect was found for ROI (*F*(2,65) = 9.431, *p* < 0.001, η^2^ = 0.199), but no main effect for Condition (*F* (3,98) = 0.695, *p* = 0.537 η^2^ = 0.018) was found. Hence, oxygenation increase differs as a function of subareas of the lPFC (ROI), but not as a function of start condition (Condition) (see Figure 3 and Table A1). A significant ROI × Time interaction (*F* (2,65) = 9.431, *p <* 0.001, η^2^ = 0.199), but no significant ROI × Condition interaction (*F* (4,168) = 0.346, *p* = 0.864, η^2^ = 0.009) was found. Thus, oxygenation differences in subareas of the lPFC (ROI) covaried with the time (Baseline vs. Set), but not with start conditions. No significant Condition × Time interaction (*F* (3,98) = 0.695, *p* = 0.537, η^2^ = 0.018) was found. Hence, start condition did not covary with time (Baseline vs. Set).

Bonferroni-corrected post-hoc pairwise comparison indicated that cortical activity significantly increased from 2 s before the Set signal to 7 s after the Set signal in anterior parts (*p* = 0.016) and ventral parts (*p* < 0.001), but not in other parts (*p* = 0.452) of the lPFC.

As we could not find a condition effect for the different sprint start sequences (no-start, action inhibition start, action initiation start, combined start), we summarized all four averaged hemodynamic changes to increase the statistical power of the subsequent analyses.

### 5.3. Regions of Interest (ROIs), Laterality and Gender Effects

We conducted a three-way mixed measures ANOVA having Gender (Male vs. Female) as a between factor and two within factors: ROI (ventral, anterior, others) to compare activity differences in specific subareas of the lPFC and Laterality (right vs. left) to compare hemispheric activity differences. Only lPFC activation 5 to 7 s after the set signal (response to set signal) was taken into account. We computed Bonferroni-corrected post hoc t-tests to assess differences between specific factor levels.

No statistically significant three-way interaction between Gender, ROI and Laterality was found, *F* (2,70) = 0.691, *p* = 0.505, η^2^ = 0.019. A large significant main effect emerged for ROI (*F* (2,70) = 14.856, *p* < 0.001, η^2^ = 0.298), but not for Gender (*F* (1,35) = 1.373, *p* = 0.249, η^2^ = 0.038) and Laterality (*F* (1,35) = 0.305, *p* = 0.584, η^2^ = 0.009). Hence, oxygenation changes were pronounced significantly stronger in specific subareas of the lPFC, but did not differ as a function of participants’ gender or between the left and right hemisphere (see Table 1 and Figure 4).

Furthermore, the three-way repeated measures ANOVA revealed a significant strong interaction Gender × ROI effect (*F* (2,70) = 10.042, *p* < 0.001, η^2^ = 0.223), but no significant Gender × Laterality (*F* (1,35) = 0.094, *p* = 0.761, η^2^ = 0.003) and ROI × Laterality (*F* (2,70) = 0.852, *p* = 0.431, η^2^ = 0.024) effect. Thus, oxygenation differences in subareas of the lPFC covaried with participants’ gender, but not with laterality.

Bonferroni-corrected post-hoc multiple pairwise comparisons indicated that oxygenation increases were significantly higher in ventral compared to anterior parts of the lPFC (*p* = 0.003) and in ventral compared to other parts of the lPFC (*p* < 0.001). For male participants, cortical activity was significantly higher in ventral parts compared to anterior parts of the lPFC (*p* < 0.001) and in ventral parts compared to other parts of the lPFC (*p* < 0.001). Thus, oxygenation changes in male participants are significantly magnified in ventral parts of the lPFC. No significant cortical activity differences were found for female participants (*p* > 0.95). Thus, lPFC oxygenation increases in female participants did not differ between subareas of the lPFC.

## 6. Discussion

In this study, we aimed to explore the cortical underpinnings of the required self-control during sprint start execution. We replicated and extended the study of Wolff et al. [43] on oxygenation changes in the lPFC during a self-control-demanding sprint start. In line with previous research, we observed a significant increase in cortical activity in the lPFC in the timeframe between the Set and Start signal of a sprint start sequence. In contrast to previous findings [43], oxygenation did not differ between experimental conditions that were designed to vary the level of task-induced self-control demands. Replicating previous research [43], the increase in lPFC oxygenation was particularly pronounced in ventral parts, whereas no oxygenation change was observed in other subareas of the lPFC. Extending previous research, we observed that the substantial increase in oxygenation in the ventral parts of the lPFC occurred in male but not in female participants. Finally, we did not find evidence for oxygenation differences between the right and left hemisphere. Taken together, our results provide further evidence for a robust involvement of the lPFC´s more ventral parts during a sprint start, but, interestingly, this effect was only found in males and it did not vary between hemispheres. We believe these findings to be important for the following three reasons.

First, our results show that only the most ventral parts of the lPFC (optodes, according to the international 5/10 system: F6–F8, F6–FC6, F5–FC5, F5–F7) responded to the self-control demands of readying oneself for an imminent sprint start with an increase in cerebral oxygenation. Herewith, we confirmed what the exploratory analyses in Wolff et al. [43] suggested. Importantly, this finding is also in accordance with research from cognitive neuroscience that different self-regulatory processes are governed by different subregions of the lPFC [39]. For example, the ventral lPFC has been shown to be responsible for response inhibition, whereas the dorsolateral PFC has been linked to top-down attention control [39], and anterior regions of the PFC have been ascribed a role in cognitive branching [62]. In the context of the self-control demands a sprint start imposes, the pronounced involvement of the ventral lPFC as the prime structure for response inhibition makes intuitive sense. Effective response inhibition in the timeframe between Set and Go might be particularly difficult for participants that do not have a background in sprint running (as is the case with the participants in this study). Indeed, research shows that prior self-control exertion increases the number of false starts in athletes with no sprint start experience [32], whereas athletes with track and field experience showed delayed starting but not more false starts if they had exerted self-control in a previous task [31]. Further tentative support for the need to control the impulse to start prematurely comes from our observation that the magnitude of the increase in ventral lPFC oxygenation did not differ between experimental conditions. This finding was surprising, as the no-start condition was expected to require less self-control demands. However, our manipulation check indicated that participants actually perceived the no start condition as being the most challenging condition. Thus, conditions that were expected to increase the demands for action initiation and/or action inhibition (action inhibition start, action initiation start, combined start) were neither perceived as being more challenging, nor did they covary with a more pronounced cortical hemodynamic response. The requirement of not starting (no-start condition) appeared to have been the most challenging.

Second, our findings shed some light on gender differences in the cortical processing of the self-control demands of a sprint start. Research from basic neuroscience points towards gender differences in the magnitude of activation changes in cortical and subcortical brain regions, such as medial frontal cingulate cortices, globus pallidus, thalamus and parahippocampal gyrus, indicating greater neural activation in men than in women [63]. In line with this, we observed a stronger oxygenation change in the ventral lPFC in men than in women. Thus, while we found robust evidence for the stronger involvement of ventral lPFC between the Set and Go signal in males, this area does not respond with a significant increase in females. One reason for our failure to observe this effect in females could be due to variations in neuroanatomy between females and males, which can lead to differences in neural activity [63] and behavioral differences [64]. However, as our fNIRS montage covered the lPFC very broadly, one might expect to see a more pronounced oxygenation increase in other channels for females. However, this was not the case. Also, in previous research, no cortical and subcortical brain regions demonstrated greater activation in women than in men during a self-control task [63]. Thus, we believe our findings are more in line with research on gender differences in regard to self-control. Indeed, research indicates that females outperform males in behavioral control tasks [44,45,65]. Importantly, females show better behavioral inhibitory control than males during a two-choice oddball task in which subjects need to respond to standard and deviant stimuli [45]. Thus, results indicate that females have advantages concerning the control of inhibitory processes. In line with this, Li et al. [63] showed that females needed less cortical and subcortical activation to achieve similar reaction times and accuracy rates in a stop signal task than males. In different words, men may require more neural resources to control their behavior. Going back to sprint starts, females might be able to deal more efficiently with the self-control demands of a sprint start. Interestingly, this interpretation is in line with data from the 2008 Bejing Olympics, where female sprinters produced only four false starts (in 387 races), compared to 25 false starts that where produced by male sprinters (in 439 races) [66].

Third, it is still an ongoing debate whether self-control demands are processed more in the right or left hemisphere or if there is no hemispheric difference in lPFC activity. Previous studies showed that self-control tasks, such as go/no-go activation, are accompanied by more activity in the left hemispheric prefrontal cortex [46,47]. Our results indicate no lateralization patterns. Cortical activity of the lPFC was not significantly higher in the right or in the left hemisphere. Also, no two-way interaction effects with gender or ROI were found. Hence, we did not find evidence for hemispheric differences in oxygenation in different subareas of the lPFC or as a function of participants’ gender.

## 7. Limitations

Against our expectations, the magnitude of cortical activation did not differ as a function of start condition. Hence, cortical activity was not significantly lower for the no-start condition compared to the other three start conditions (promotion, prevention, optimal). Thus, manipulations that supposedly added different self-control demands (impulse control and action initiation) did not lead to higher cortical activation in contrast to a singular self-control demand (impulse control). These findings do not support our hypothesis and previous conclusions [43]. Possibly, our manipulation was not successful, as indicated by the manipulation check: The no-start condition was perceived as the most difficult one. This was unexpected because compared to the other three sprint start conditions, the no-start condition only imposes impulse control (to not start). As the application of self-control produces costs and is invested sparingly [22,23,24], less complex self-control demands (only impulse control) should be perceived as less aversive and difficult. However, given that responding to the go signal might be a very strong behavioral impulse, it is conceivable that the intensity of the required control signal needs to be comparably high to control this almost automatic impulse.

Perceived subjective difficulty is frequently used as a marker to assess perceived self-control investment and costs. Maybe more individually challenging start conditions might lead to higher cortical activation and higher demand of self-control. Therefore, subjective difficulty of sprint start conditions need to be measured in a more differentiated way (e.g., 7 point Likert-Scale), to allow for a comparison between conditions in more absolute terms. Additionally, physical and psychological variables like perceived effort, frustration, fatigue or cognitive overload might also contribute to the processing of control demands, and it would be interesting to also assess such variables. However, our manipulation check showed that participants were generally motivated to perform well and were not extremely exhausted.

## 8. Future Research

In the current study, we did not examine to what extent activity in the ventral lPFC is related to the actual start performance. Evaluating behavioral performance in a sprint start task could be operationalized by measuring the reaction time. McEwan, Ginis & Bray [67] demonstrated in a dart-throw task that effective self-control increases reaction time of movement initiation and accuracy of throwing motion. More importantly, Englert et al. [31,32] showed that participants who had completed a self-control-demanding task prior to a sprint start displayed worse reaction times. We believe it would be an important question for future research to unravel the relationship between the cortical response after the Set signal and the resultant reaction time.

Differences in behavioral performance (here: reaction time) can also arise in consequence to experience differences: elite sprinters differentiate from well-trained sprinters due to significantly faster starts in 60 and 100 m races [12]. Englert et al. [32] showed that prior self-control exertion leads to worse starting times in elite sprinters, but no increase in the number of false starts, whereas the number of false starts increases in athletes with no sprint start experiences. Linking these findings, it is highly interesting and promising for future research to investigate oxygenation and behavioral differences (e.g., reaction time, false start rate and start technique) between novice and professional sprinters. Referring to the findings of Lipps et al. [66], which show a lower false start rate for women than for men, it is a fascinating research question to compare these behavioral differences between female and male sprinters and assess how they covary with changes in lPFC activation changes. Further variables of interest for future research are how difficult participants perceive the task to be and how they feel during the testing session. Thus, it might be instructive to assess these variables with the Profile of Mood States (POMS) [68] questionnaire and Borg Scale [69] before and after the sprint start in future studies.

## 9. Conclusions

It is well established that self-control is indispensable for an optimal sport performance, such as sprint start execution [31,32,43]. In line with previous research, we observed a significant increase in cortical activity in the lPFC during sprint start preparation. Additionally, we showed that particularly ventral parts of the lPFC are activated prior to the Go signal of a sprint start. Hence, exploratory findings [43] concerning the importance of ventral parts of lPFC for self-control execution during sprint starts could be replicated. In extension to this, we observed that a significant increase in cortical activity in the ventral parts of the lPFC did not occur in females, but only in male participants. No lateralization patterns were found. This study transfers findings from basic neuroscience to sports and facilitates the understanding of the cortical processing of sport-specific self-control demands.

## Figures and Tables

**Figure 1 brainsci-10-00494-f001:**
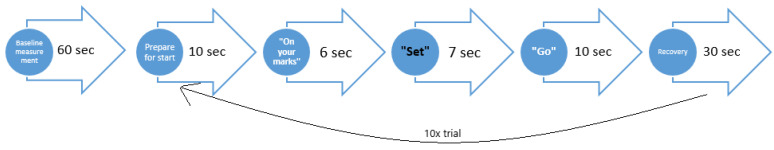
Procedure of a sprint start sequence. Structure and timeframe are the same for all four sprint start conditions. Conditions differ only by task instruction.

**Figure 2 brainsci-10-00494-f002:**
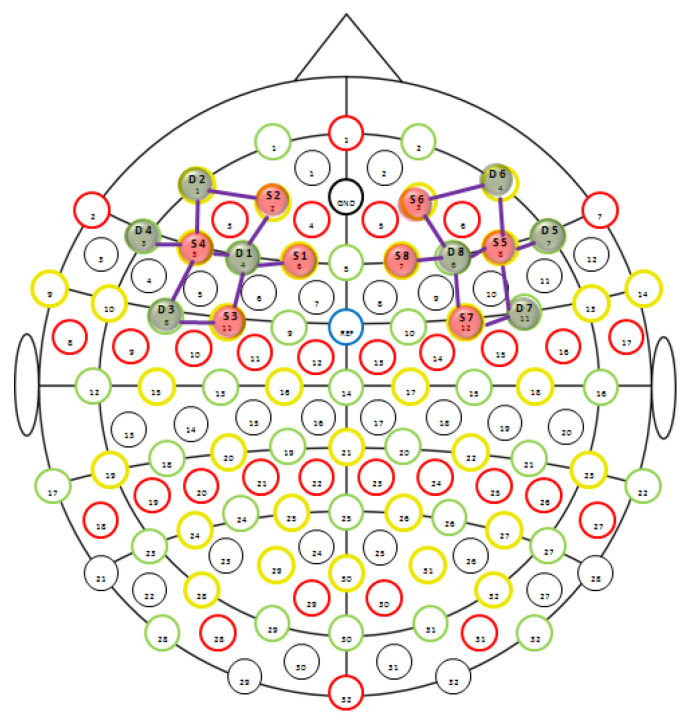
8 Sources and 8 detectors were placed according to the international 5/10 system: E1 at F1, E2 at AF3, E3 at FC3, E4 at F5, D1 at F3, D2 at AF7, D3 at FC5, D4 at F7, E5 at F6, E6 at AF4, E7 at FC4, E8 at F2, D5 at F8, D6 at AF8, D7 at FC6, and D8 at F4.

**Figure 3 brainsci-10-00494-f003:**
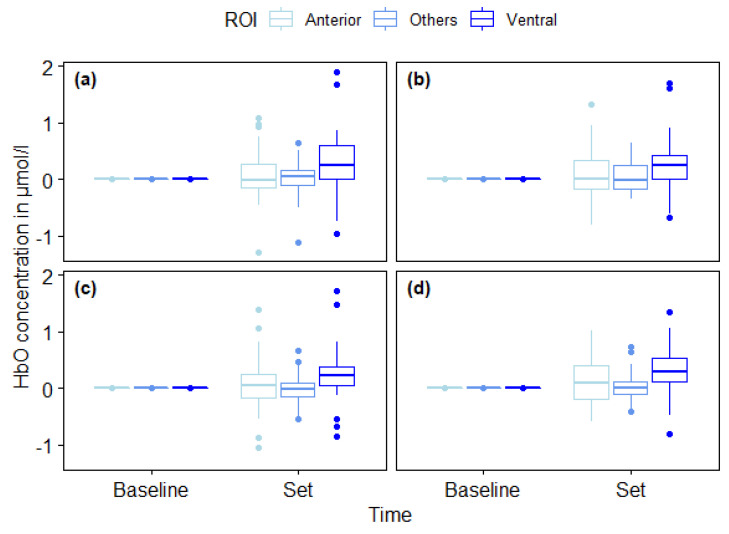
Boxplots of oxygenated hemoglobin (HbO) concentrations (in μmol/L) grouped in regions of interest (ROIs) (anterior, ventral, others) for all four sprint start conditions (action inhibition start (**a**), action initiation start (**b**), combined start (**c**), no-start (**d**)) 2 s before the set signal (baseline) and 5 to 7 s after the set signal (response to set signal). Middle lines inside each box represent the median scores, top lines of each box indicate the upper quartiles, bottom lines of each box indicate lower quartiles, upper and lower vertical lines indicate upper and lower whiskers and dots represent outliers.

**Figure 4 brainsci-10-00494-f004:**
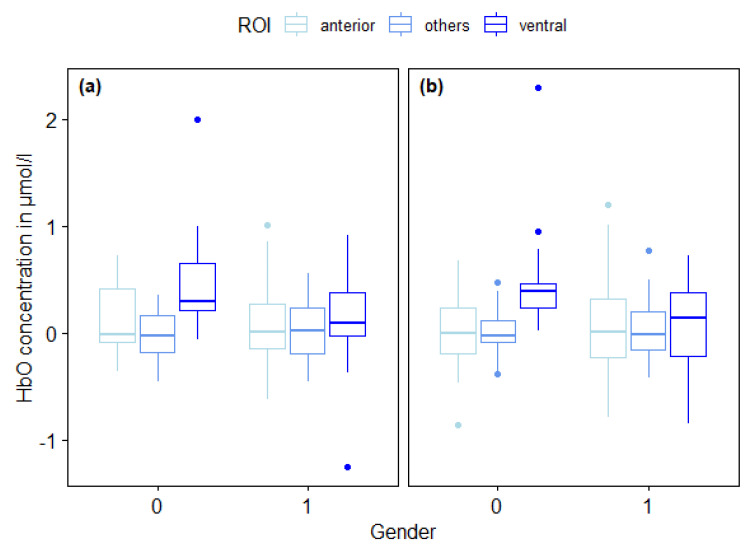
Boxplots of group averaged oxygenated hemoglobin (HbO) concentrations (in μmol/L) 5 to 7 s after set signal differentiated for regions of interest (ROIs) (anterior, others, ventral), Gender (Male = 0, Female = 1) and for Laterality (left (**a**) and right (**b**) hemisphere). Middle lines inside each box represent the median scores, top lines of each box indicate the upper quartiles, bottom lines of each box indicate lower quartiles, upper and lower vertical lines indicate upper and lower whiskers and dots represent outliers.

**Table 1 brainsci-10-00494-t001:** Values of oxygenated hemoglobin (HbO) concentration.

Regions of Interest (ROIs)	Laterality	Male		Female		Mean Differences
		*n*	M (SD)	*n*	M (SD)	
Ventral	Right Hemisphere	17	0.492 (0.514)	21	0.070 (0.425)	0.422
	Left Hemisphere	17	0.503 (0.485)	21	0.135 (0.432)	0.368
Anterior	Right Hemisphere	16	0.014 (0.396)	22	0.060 (0.496)	−0.046
	Left Hemisphere	17	0.147 (0.345)	22	0.083 (0.379)	0.064
Others	Right Hemisphere	17	0.004 (0.243)	22	0.035 (0.300)	−0.031
	Left Hemisphere	17	−0.024 (0.237)	22	0.027 (0.288)	−0.051

Note. *n* = number of persons; M = mean value; SD = standard deviation.

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
