# Peer review of "On Your Mark, Get Set, Self-Control, Go: A Differentiated View on the Cortical Hemodynamics of Self-Control during Sprint Start"

_brainsci, 2020, doi:10.3390/brainsci10080494_

Round 1

Reviewer 1 Report

Thank you for asking me to review the manuscript titled, “On your mark, get set, self-control, go: A differentiated view on the cortical hemodynamics of self-control during sprint start.” The following are my comments regarding the manuscript, section by section.

Title:

  • The title appropriately describes the study.

Abstract:

  • Line 18 - I suggest changing “within-design” to “within-subject design.”
  • Otherwise, the abstract is well-written and informative.

Introduction:

  • Line 32 - change starts to start.
  • Line 56 - “…that would be required.” An example here would be useful as the sentence doesn’t seem complete.
  • Perhaps it would be useful to provide evidence in clinical populations, such as ADHD, that frontal activation (IPFC) is associated with decreased inhibition and attention.
  • Otherwise, the Introduction is well-written, informative, and interesting.

Methods:

  • Line 148 - what time were the study sessions conducted? Morning? Afternoon? Whenever they were available? This should be stated.
  • Otherwise the fNIRS methodology appears to be valid and is appropriately described. Well done.

Results:

  • Line 246 - were the six participants who did not respond to the manipulation check question included in subsequent analyses?
  • Otherwise, the results are very well written. I have no further comment - great job.

Discussion:

  • Line 378 - this is great behavioral evidence to support your claim! I was hoping this would be portrayed in the manuscript.

This is a very well-written manuscript. Hence, I do not have many comments. I wish the authors all the best with their future endeavors and hope to review more manuscripts from them in the future.

Author Response

Dear Reviewer I,
Thank you for the constructive review concerning our paper entitled “"On your mark, get set, self-control, go: A differentiated view on the cortical hemodynamics of self-control during sprint start” submitted to the Special Issue “Studying Brain Activity in Sports Performance" of Brain Science.
We addressed the remarks and have tried to change the manuscript accordingly (for detailed replies to each issue that was raised, please see the table in the attachment). All revisions are clearly highlighted using the “Track Changes” function.
We again thank you for your constructive review. By incorporating your suggestions, we are confident that the changes we have made to the original version of the paper have improved its quality substantially.

Sincerely,
Kim-Marie Stadler, Wanja Wolff & Julia Schüler

Reviewer 2 Report

Dear authors,

The revised paper is an important contribution to the field of Sports Psychology and Sports Science. However, the reviewer recommends making some modifications to the article to achieve the quality standards required by the Journal.

The reviewer particularly values studies with quasi-experimental designs and recommends that the authors continue in the same line of work.

Regards

Author Response

Dear Reviewer 2,
Thank you for the constructive review concerning our paper entitled “"On your mark, get set, self-control, go: A differentiated view on the cortical hemodynamics of self-control during sprint start” submitted to the Special Issue “Studying Brain Activity in Sports Performance" of Brain Science.
We addressed the remarks and have tried to change the manuscript accordingly (for detailed replies to each issue that was raised, please see the table in the attachment). All revisions are clearly highlighted using the “Track Changes” function.
We again thank you for your constructive review. By incorporating your suggestions, we are confident that the changes we have made to the original version of the paper have improved its quality substantially.

Sincerely,
Kim-Marie Stadler, Wanja Wolff & Julia Schüler

Reviewer 3 Report

I find it a very interesting and well-executed study. The development of the paper is correct and of high quality. I congratulate the authors. 

Introduction

The introduction is fine. Congratulations. Although I would like to suggest the following: the first paragraph starts like this: “Winning or losing in a sprint race is determined by milliseconds and highly dependent on a perfect start (Brown & Vescovi, 2012; Helmick, 2003; Majumdar & Robergs, 2011; Salo & Bezodis, 2004). The sprint starts contributes 5% of the overall 100m race time (Harland & Steele, 1979).” So, are lines 33-37 necessary? I mean, wouldn't it be better to continue talking specifically about sprint race? Perhaps it is not pertinent to speak of sport in general, when the development of the article deals with such a specific field. 

Otherwise, I like the introduction. The object of study is focused, adequately supported and highlights why study is important.

Methods

Authors indicate that:…”who had no further experience in sprint starts but did sport regularly”. I suggest that more information be given on the characteristics of the participants' physical activity: type of physical activity, frequency of practice, etc.

I suggest explaining the inclusion and / or exclusion criteria of the participants.

If authors have the data, I suggest that they include anthropometric and physical fitness characteristics of the participants.

Please indicate if any ethical committee endorsed the study.

Results

Indicate if the data meets the normality criteria to use ANOVA.

I suggest that the authors decide whether to use 2 or 3 decimals in the value of the statistics.

Discussion

I suggest starting the discussion by explaining the research objective.

I consider that the discussion is very well constructed and the results found are well justified.

Author Response

Dear Reviewer 3,
Thank you for the constructive review concerning our paper entitled “"On your mark, get set, self-control, go: A differentiated view on the cortical hemodynamics of self-control during sprint start” submitted to the Special Issue “Studying Brain Activity in Sports Performance" of Brain Science.
We addressed the remarks and have tried to change the manuscript accordingly (for detailed replies to each issue that was raised, please see the table in the attachment). All revisions are clearly highlighted using the “Track Changes” function.
We again thank you for your constructive review. By incorporating your suggestions, we are confident that the changes we have made to the original version of the paper have improved its quality substantially.

Sincerely,
Kim-Marie Stadler, Wanja Wolff & Julia Schüler
